# Thermo-Responsive Hydrophilic Support for Polyamide Thin-Film Composite Membranes with Competitive Nanofiltration Performance

**DOI:** 10.3390/polym14163376

**Published:** 2022-08-18

**Authors:** Haniyeh Najafvand Drikvand, Mitra Golgoli, Masoumeh Zargar, Mathias Ulbricht, Siamak Nejati, Yaghoub Mansourpanah

**Affiliations:** 1Membrane Research Laboratory, Lorestan University, Khorramabad 68151-44316, Iran; 2School of Engineering, Edith Cowan University, Joondalup, WA 6027, Australia; 3Lehrstuhl für Technische Chemie II and Center for Water and Environmental Research (ZWU), Universität Duisburg-Essen, 45117 Essen, Germany; 4Department of Chemical and Biomolecular Engineering, University of Nebraska-Lincoln, Lincoln, NE 68588, USA

**Keywords:** poly(N-isopropylacrylamide) (PNIPAAm), thermo-responsive membrane, hydrophilic hydrogel support, interfacial polymerization

## Abstract

Poly(N-isopropylacrylamide) (PNIPAAm) was introduced into a polyethylene terephthalate (PET) nonwoven fabric to develop novel support for polyamide (PA) thin-film composite (TFC) membranes without using a microporous support layer. First, temperature-responsive PNIPAAm hydrogel was prepared by reactive pore-filling to adjust the pore size of non-woven fabric, creating hydrophilic support. The developed PET-based support was then used to fabricate PA TFC membranes via interfacial polymerization. SEM–EDX and AFM results confirmed the successful fabrication of hydrogel-integrated non-woven fabric and PA TFC membranes. The newly developed PA TFC membrane demonstrated an average water permeability of 1 L/m^2^ h bar, and an NaCl rejection of 47.0% at a low operating pressure of 1 bar. The thermo-responsive property of the prepared membrane was studied by measuring the water contact angle (WCA) below and above the lower critical solution temperature (LCST) of the PNIPAAm hydrogel. Results proved the thermo-responsive behavior of the prepared hydrogel-filled PET-supported PA TFC membrane and the ability to tune the membrane flux by changing the operating temperature was confirmed. Overall, this study provides a novel method to fabricate TFC membranes and helps to better understand the influence of the support layer on the separation performance of TFC membranes.

## 1. Introduction

With the rapid growth of the world’s population and water contamination, the universal need for freshwater has increased more rapidly than in the past [1]. Along these lines, membrane technology is becoming one of the most effective strategies to purify water and wastewater and produce freshwater [2,3,4,5]. Depending on the membranes’ pore size and rejection mechanism in pressure-driven processes, they are classified into microfiltration, ultrafiltration, nanofiltration, and reverse osmosis membranes [6]. Among different membranes used in water purification, thin-film composite (TFC) membranes have been widely used in the nanofiltration and reverse osmosis process due to their high selectivity, tunable structure, and chemical and thermal stability [7]. The TFC membranes are composed of a dense polyamide (PA) active layer supported by porous polymeric layers. The microporous layers, immediately in contact with the PA active layer, are normally prepared via the phase inversion technique, and interfacial polymerization is the dominant method to prepare the active PA layer on the surface of the support [8]. This method enables independent optimization of the support layer and the active layer to achieve significant selectivity and permeability [9]. Although researchers have mostly focused on PA layer modification, there are a few studies that investigated the impact of the support layer properties on the TFC membranes’ performance [10,11,12]. The porous support layer mainly provides the required mechanical strength; however, its properties such as hydrophobicity, porosity, pore size, and roughness influence the formation of the PA layer [11,12]. Polysulfone (PSF) and polyethersulfone (PES) are mainly used as TFC support layers owing to their relatively high thermal resistance, chemical stability, and easy fabrication [13,14,15,16]. Usually, the PSF or PES support membrane is cast on a PET non-woven support to provide the desired mechanical strength. The main disadvantage of PSF and PES membranes is that they are hydrophobic whereas hydrophilic support layers facilitate the interfacial polymerization reaction and PA selective film formation and can ultimately result in higher permeability and performance of the developed TFC membranes [17,18,19]. Many studies have explored the hydrophilic modification of the conventional support layers to enhance the performance of TFC membranes [20,21]. However, modification methods commonly require a complicated or harsh additional step to achieve the desired properties [9,22]. Hence, developing new hydrophilic support layers could make a breakthrough in designing highly efficient TFC membranes [23].

Recently, stimuli-responsive hydrogels have gained increasing attention in membrane fabrication owing to their ability to control their properties in response to environmental changes (e.g., pH, temperature, light) [6,24,25,26]. Hydrogels are three-dimensional network structures composed of polymeric chains that can absorb water, undergo significant volume expansion (swelling), and form a hydration layer on their surfaces [27,28]. These characteristics make them promising candidates for integration into membranes for enhanced anti-fouling properties [6,28]. For instance, Zhang et al. [29] prepared a modified PES membrane by grafting a novel polyampholyte hydrogel onto the membrane surface and achieved low fouling and high flux recovery of the modified membrane due to high hydration of the grafted hydrogel. Additionally, introducing hydrogel into the architecture of membranes has two important features. First, the three-dimensional network of hydrogels can be considered microscopically as a porous structure [30,31,32]. Second, the sieving coefficient as a function of hydrogel mesh size may be tuned by an external stimulus such as pH or temperature. For instance, membranes with grafted thermo-responsive hydrogels are able to alter their pore size and surface properties by changing temperature; this attribute makes separation efficiency tunable [33,34].

PNIPAAm is a common thermo-responsive polymer that demonstrates a lower critical solution temperature (LCST) at 32 °C [22,35,36]. A porous membrane that is grafted with PNIPAAm reduces the membrane pore size at temperatures below 32 °C (LCST) due to swelling, and hydrogel dehydration and collapse lead to pore opening above 32 °C [22,37]; this enables control over the water permeation efficiency of the membranes by temperature alteration. Pressure-driven mass transfer through swollen hydrogels is only possible when the gel is stabilized in a porous support matrix to maintain its integrity [22,38,39]. For example, Adrus and Ulbricht [40] used a polyethylene terephthalate (PET) support for developing a hydrogel pore-filled microfiltration membrane. The PNIPAAm was grafted into the PET structure altering the accessible pore volume while not changing the overall membrane thickness. The developed microfiltration membrane showed size-selective barrier properties in response to the temperature due to swelling and deswelling of PNIPAAm, confirming its thermo-responsive behavior. To the best of the authors’ knowledge, the incorporation of thermo-responsive hydrogels into the PET support layer of TFC membranes has never been explored. PNIPAAm can be a promising hydrogel to conduct a proof of concept study due to its hydrophilicity and thermo-responsive characteristics. A PET non-woven that is pore-filled with PNIPAAm can replace the microporous support layer of TFC membranes, thereby reducing the overall thickness of the TFC structure, which can also rectify the high energy demand associated with thick support layers [12].

Here, we report on a thermo-responsive TFC membrane prepared by interfacial polymerization of PA active layer on a hydrogel-filled PET support scaffold without the application of a microporous membrane interlayer. To integrate PNIPAAm within non-woven PET and create a hydrophilic substrate, we used a reactive pore filling approach. The fabricated hydrogel support and PA TFC membrane (with no microporous support layer) were analyzed using different characterization techniques to confirm the successful formation of the PA layer on the developed novel support. Water permeability and salt rejection of the developed membranes were explored. Finally, the separation performance was measured below and above the LCST of PNIPAAm to investigate the thermo-responsive behavior of the membranes.

## 2. Materials and Methods

### 2.1. Materials

N-isopropylacrylamide (NIPAAm) (97%) was purchased from Sigma (Taufkirchen, Germany). Tetramethylethylenediamine (TEMED), ammonium persulfate (APS), N,N′-methylenebisacrylamide (MBA), 1,3,5-benzenetricarbonyltrichloride (TMC, 98%), n-hexane, p-phenylenediamine (PPD, 99%), triethylamine (TEA, 99.5%), polyethylene glycol (PEG-600), benzophenone, sodium chloride (NaCl,) propanol, and ethanol were obtained from Merck (Steinheim am Albuch, Germany).

### 2.2. Membrane Fabrication

#### 2.2.1. Fabrication of a Novel Hydrogel Support

First, the voids of the non-woven PET scaffold were filled with hydrogel. For this purpose, the non-woven PET fabric was immersed in a solution of 0.1 M benzophenone in an ethanol–water mixture with a ratio of 10:1 (*v*/*v*%) and allowed to rest for 2 h. After adsorption of the photo-initiator, the non-woven fabric was placed in a Petri dish containing 0.2 M aqueous solution of NIPAAm and was irradiated with a UV source (160 W) for 15 min. The non-woven PET was then placed in distilled water for 24 h to remove the unreacted materials and was dried at room temperature [40,41]. For the reactive pore-filling, a solution containing 0.9 g of NIPAAm, 0.018 g of MBA as the cross-linker, and 270 µL of APS (10 wt.%) as the initiator in 10.8 mL of water was prepared. The PET scaffolds were inserted into a filter holder and the solution was continuously circulated for 1 h (Figure 1). The membrane was then removed and placed in a beaker containing the same solution and stirred for 1 h at 250 rpm at room temperature. Thereafter, 20 µL of TEMED as a promoter of the initiator APS for hydrogel formation was added to the solution, which was stirred for another 30 s. The membrane was then placed between two glass plates for 24 h for the cross-linking polymerization reaction to occur. Finally, the prepared membrane (noted as PET-PNIPAAm membrane) was placed in distilled water for 24 h to remove unreacted materials [39,41,42].

#### 2.2.2. Fabrication of PA TFC Membrane

A PA thin layer was fabricated by the interfacial polymerization (IP) method, in which active amine monomers are allowed to react with organic monomers at the interface between the aqueous and organic phases to form a network structure [8]. To do so, the prepared support was placed in a frame with a 2.2 cm diameter and 1.0 cm depth and an aqueous solution containing 0.4 wt.% PPD and 0.8 wt.% TEA was poured on the membrane. The solution was allowed to remain for 5 min within the frame before being drained at room temperature. The membrane surface was pressed by a soft rubber roller to remove the extra solution. The organic solution (TMC, 0.2 wt.% in n-hexane) was poured on the membrane and kept for 3 min. The excess organic solution was then removed from the frame, and the prepared membrane was placed in an oven at 70 °C for 30 min to dry and cure. The resulting TFC membrane was finally washed with a hexane-propanol mixture with a ratio of 3:1. Finally, the as-prepared composite (noted as PET-PNIPAAm-PA membrane) was kept in DI water at 4 °C until further characterization and performance evaluations.

#### 2.2.3. Modified Hydrogel Support

PEG as a pore-forming agent was used to improve the membrane water permeability. The method for the preparation of hydrogels modified with PEG was the same as the plain membrane fabrication noted in Section 2.2.1; the only difference was the addition of 1.8 g PEG (as a pore-forming agent in the hydrogel network) to the same composition of the aqueous solution. The membrane prepared in this step is noted as a PET-PEG-PNIPAAm membrane. The PA layer was fabricated on the modified membrane according to the method noted in Section 2.2.2 and the final membrane is referred to as the PET-PEG-PNIPAAm-PA membrane.

### 2.3. Membrane Characterization

#### 2.3.1. Scanning Electron Microscopy (SEM) and Energy Dispersive X-ray Spectroscopy (EDX)

To investigate the hydrogel loading of pore-filled polyester and the formation of PA, an FE-SEM (TESCAN, Brno, Czech Republic) coupled with energy-dispersive X-ray spectroscopy (EDX) was used. Membrane samples were freeze-fractured in liquid nitrogen, sputter coated with gold, and mounted on SEM stubs. The non-woven hydrogel membrane sample was dried using a freeze dryer for better observation of the structure of hydrogels. The surface and cross-section of membranes were observed by SEM at 15.0 kV. The elemental composition information of the samples was obtained by EDX accordingly.

#### 2.3.2. Atomic Force Microscopy (AFM)

AFM device (DME model C-21, Copenhagen, Denmark) was used to determine the surface roughness and morphology of the membranes. Small pieces of prepared membrane samples (1 cm^2^) were cut and attached to a glass plate using double-sided tape; all samples were air-dried overnight before AFM analysis. The membrane samples were scanned and observed in a non-contact mode in the air at room temperature with a silicon probe in a scan size of 2 μm × 2 μm. SPM-DME software was used to measure roughness values. Three measurements were done to calculate the roughness for each sample, and average values are reported.

#### 2.3.3. Water Contact Angle (WCA)

The hydrophilicity of the membranes was determined by the contact angle device (G10, KRÜSS Co., Nürnberg, Germany). In all measurements, a water droplet of 1 μL was placed on the membrane surface. Furthermore, the water contact angle was also determined at both room temperature and at 45 °C to investigate the hydrophilicity–hydrophobicity behavior of membranes upon temperature change. A heater was placed under the sample to evaluate the membrane hydrophilicity at different temperatures while the temperature was equilibrated for 10 min before measuring the contact angles. For all samples, at least five measurements at different locations of the sample were performed and the average is reported.

### 2.4. Membrane Separation Performance Evaluation

A dead-end filtration unit was used to evaluate the performance of membranes. The permeate flux of all membrane samples was measured for 90 min at 1 bar. The permeate flux (J) and water permeability (A) were calculated by measuring the changes in the volume of the permeate over time using Equations (1) and (2), respectively:J = ∆V/(Am ∆t) (L/m^2^ h)(1)
A = J/∆P (L/m^2^ h bar)(2)
where Am is the specific surface area of the membrane, Δt is the measurement time and ΔP is the applied pressure.

To perform the membrane rejection tests, a 1000 ppm solution of NaCl was prepared as a feed solution. The salt concentration of permeate was evaluated with a conductivity meter using a calibration curve established for NaCl. Finally, the rejection of NaCl was calculated using Equation (3):R (%) =1 − C_p_⁄C_0_(3)
where C_p_ is the concentration of the permeate and C_0_ is the feed concentration. At least three tests were done to investigate the salt rejection performance for each salt and the average values are reported.

### 2.5. Effect of Temperature on the Performance of the Novel PA TFC Membrane

To investigate the effect of temperature, the hydrogel support was prepared according to Section 2.2.1 and then dried at 70 °C for 15 min to reach a hydrophobic state. The new non-woven hydrogel membrane was then used for PA layer fabrication according to Section 2.2.2 and is noted as PET-hydrophobic PNIPAAm-PA. Finally, to investigate the effect of temperature on membrane performance, the water permeability and salt rejection tests were performed at both room temperature and at 45 °C.

## 3. Results and Discussions

### 3.1. Membrane Characterization

#### 3.1.1. SEM, AFM, and EDX

The surfaces of PET, PET-PNIPAAm, and PET-PNIPAAm-PA membranes were analyzed by SEM and AFM to evaluate their surface topography and surface morphology. Surface and cross-sectional SEM images as well as their AFM images are presented in Figure 2. The results show that the PET scaffold consists of fibers that provide enough voids for hydrogel loading between them (Figure 2a). Figure 2b corresponds to the PET-PNIPAAm membrane and confirms PNIPAAm hydrogel integration inside the non-woven fabric, filling the empty space between the fibers. Finally, Figure 2c shows a thin layer of PA covering the PET-PNIPAAm membrane and confirms the formation of PA ridge and valley structure on the hydrogel pore-filled non-woven membrane in both surface and cross-section SEM images. The results affirm the suitability of the proposed technique to develop a hydrogel-reinforced support layer for TFC membranes and its suitability for PA formation.

The average roughness (R_a_), root mean square roughness (R_q_), and maximum roughness (R_z_) values are reported in Table 1. PET, PET-PNIPAAm, and PET-PNIPAAm-PA membranes showed roughness (R_q_) of 24, 16, and 16 nm, respectively. The distinctive topography of the pristine non-woven PET membrane is shown in the AFM image (Figure 2a) with the light regions representing the highest points and the dark regions being the depth of the valleys. The results confirm that the PET backbone has a relatively rough surface compared to the PET-PNIPAAm membrane due to its non-uniform non-woven fiber structure. The topography of the PET textiles after the hydrogel grafting is shown in Figure 2b, implying the smooth surface of the PET-PNIPAAm membrane that is in accordance with the SEM result. The surface displayed a less pronounced peak to valley topography leading to a decreased roughness (cf. Table 1). This trend is consistent with the literature [43,44]. For instance, Kurşun et al. [45] developed a thermo-responsive membrane by grafting PNIPAAm on a poly(vinyl alcohol) membrane. They reported a smoother surface after PNIPAAm integration on a polymeric surface and correlated that to the PNIPAAm’s presence. Hence, the surface morphological changes of the PET-PNIPAAm membrane suggest the successful integration of the hydrogel layer with the PET fabric. After the interfacial polymerization and PA formation on the hydrogel membrane (PET-PNIPAAm-PA membrane), the roughness was slightly increased compared to the PET-PNIPAAm membrane surface (cf. Table 1), which can be attributed to the formation of PA ridge and valley structure [46]. Therefore, the roughness data confirmed the successful formation of non-woven hydrogel and PA layer on top of it.

The EDX detector on the SEM instrument was used to evaluate the fraction of elements (i.e., carbon, oxygen, and nitrogen here). The results are presented in Table 2. The PET scaffold showed an oxygen content of 34.8 wt.% and carbon content of 65.2 wt.% which is consistent with the value expected for the structure of the material [47]. The nitrogen element emerged in the PET-PNIPAAm membrane (13.8 wt.%), which is correlated to the presence of the amide group after pore-filling of PNIPAAm hydrogel in PET non-woven [48]. The fraction of nitrogen was further increased from 13.8 to over 24 wt.% upon PA formation in the PET-PNIPAAm-PA membrane which is correlated to the nitrogen-rich structure of the polyamide active layer. The results confirm the successful hydrogel integration in the PET non-woven and the PA formation on top of the PET-PNIPAAm membrane.

The structural difference between the conventional PES-supported PA thin layer compared with the PET-PNIPAAm-based thin PA layer is clearly shown in Figure 3. The second one clearly depicts a relatively rougher skin layer structure. The main reason for such a difference can be found in the impact of the high hydrophilic properties of PET-PNIPAAm support. PNIPAAm effectively enhanced the hydrophilicity of the backbone support which impacted the diffusion rate of the amine PPD toward the organic phase during the interfacial polymerization reaction. Hence, a very thin PA layer with a denser structure in conjunction with less surface defect has been fabricated [49,50].

#### 3.1.2. Water Contact Angle

The surface hydrophilicity of the membranes as well as their potential ability to change between more hydrophobic and more hydrophilic characteristics due to the temperature change was determined through water contact angle measurement [48,51]. These were done by measuring the contact angle of a static water drop at room temperature (23 °C) and 45 °C on membrane samples using a sessile drop technique with a tensiometer (Figure 4). This temperature range covers values that are higher and lower compared to the LCST, to account for thermo-responsive characteristics of the hydrogel membrane. WCA results at 23 °C show that the integration of the PNIPAAm hydrogel within the fabric as a pore filler decreased the contact angle of the PET scaffold from 57° to 23°, indicating that the membranes are more hydrophilic which is attributed to the incorporation of the hydrophilic functional groups of the hydrogel PNIPAAm [52,53]. After the thin PA layer formation on the PET-PNIPAAm membrane, the contact angle value increased compared to the PET-PNIPAAm and reached 41°. This can be attributed to the aromatic characteristics of the dense PA layer leading to reduced hydrophilicity of the PET-PNIPAAm-PA membrane compared to its PET-PNIPAAm precursor [54]. WCA results are also in accordance with the roughness values of membranes as increased roughness correlates with increased hydrophobicity [55,56].

WCA results at 45 °C indicate that by increasing the temperature, no significant change in the contact angle of the PET scaffold was observed. However, the contact angle value for the PET-PNIPAAm membrane increased from 23° to 72°, showing the pronounced thermo-responsive behavior of the fabricated membrane [6,43,57,58,59,60]. PNIPAAm hydrogel has an insoluble three-dimensional network structure that has reversible swelling properties in water. An increase in temperature causes the alteration of PNIPAAm chains’ structure, leading to a pronounced deswelling, i.e., the release of water, and exposure of hydrophobic isopropyl groups on the membrane surface. This results in an increase in the WCA value [6]. Modigunta et al. [48] also investigated the WCA of a porous honeycomb-patterned polystyrene film integrated with PNIPAAm below and above the LCST and reported higher WCA above the LCST. In addition, the WCA value of the PA membrane at 45 °C was higher than its WCA at 23 °C. The PA layer itself is not responsive to temperature; therefore, this change is due to the effect of the hydrogel used as a support. This indicates that the PET-PNIPAAm membrane can preserve to some extent its thermo-responsive characteristic upon PA formation on its surface. This is a surprising finding because the SEM and AFM results indicate that the PA film should cover the support completely. More insights into the barrier function of the PA film will be obtained from the nanofiltration studies (see Section 3.2 and Section 3.3).

### 3.2. Membrane Separation Performance Evaluation

The flux and salt rejection of PET, PET-PNIPAAm, and PET-PNIPAAm-PA membranes were investigated to determine their separation ability and flux performance. As shown in Table 3, the water permeability of the prepared membranes was sharply reduced compared to the PET non-woven. Water permeability decreased from 16,920 (L/m^2^ h bar) for PET scaffold to 1.5 (L/m^2^ h bar) for PET-PNIPAAm membrane because the hydrogel was able to fill the empty spaces of the non-woven well, whereas the fabrication of the PA layer in the next step caused a further water permeability reduction to 1.0 (L/m^2^ h bar) due to the compact structure of the PA layer [46]. The reactive pore filling has been performed by a cross-linking copolymerization of the PNIPAAm hydrogel. The tight entanglement ensures that the composite is stable in water. This had also been demonstrated in earlier work by Adrus and Ulbricht [40], where almost the same methodology had been used for reactive pore-filling of PET track-etched membranes that had thereafter been proven to be stable ultrafiltration membranes.

Although the hydrogel has sieving properties due to its polymer network mesh structure, the PET-PNIPAAm membrane did not show any rejection of NaCl. This is reasonable considering the mesh size in the range of several nm [40] and the uncharged structure of the polymer. However, after the interfacial polymerization, the PET-PNIPAAm-PA membrane on this support yielded 33.8% rejection against NaCl. This demonstrates a good salt selectivity compared to reported NaCl rejections of 10–20% for two industrial PA TFC nanofiltration membranes [61]. After the modification of the PET-PNIPAAm membrane by the integration of PEG as a pore-forming agent during the cross-linking polymerization, the performance of the modified hydrogel membrane (PET-PEG-PNIPAAm) and PA TFC membrane fabricated on the modified support (PET-PEG-PNIPAAm-PA) was investigated to evaluate the impact of the modified hydrogel structure. The water flux of the fabricated membranes and their NaCl rejection are presented in Table 3. Resulting from the application of PEG as a pore-forming agent in the structure of the hydrogel, the membrane flux changed from 1.5 L/m^2^ h (for PET-PNIPAAm membrane) to 37.4 L/m^2^ h (for PET-PEG-PNIPAAm- membrane). It is known that the addition of PEG into the polymerization mixture leads to the formation of a porous, phase-separated PNIPAAm hydrogel compared to a homogenous one without PEG [62]. This is the reason for the much larger permeability. However, the permeate flux of the PET-PEG-PNIPAAm-PA membrane did not show much difference compared with the PET-PNIPAAm-PA membrane. This is not very surprising when considering that the PA barrier film has the largest contribution to the resistance of a PA TFC membrane. A similar result was reported by Jimenez-Solomon et al. [63] that used PEG for modification of PA TFC support where the modified support membrane did not impact the flux of PA TFC prepared on it. Despite that, the NaCl rejection increased from 34% for the PET-PNIPAAm-PA membrane to 47% for the PET-PEG-PNIPAAm-PA membrane. This could be due to the higher chance of PA formation within the pores of the support layer due to the PPD monomers penetration in the larger pores generated through PEG influence onto the formation of a macroporous instead of a homogeneous hydrogel [12,64].

In addition, considering the low operating pressure of 1 bar used in this study, the developed novel TFC membrane showed a good balance between salt rejection and water permeability in comparison with different TFC nanofiltration membranes reported in the literature (Table 4). Overall, this study was able to achieve a suitable efficiency by reducing energy consumption by using low operating pressure.

### 3.3. The Effect of Temperature on Membrane Performance

The fabrication of a thermo-responsive TFC membrane may lead to the development of controllable and more efficient processes. For example, the efficiency of thermo-responsive TFC membrane in desalination and wastewater treatment can be designed according to the seasonal changes in temperature or using external waste heat in cases where the increase in temperature can improve membrane performance [68]. The effect of temperature on the developed hydrogel pore-filled non-woven was first investigated by drying hydrogel after fabrication which caused the membrane to be in a hydrophobic state [6]. This caused lower water permeability after PA fabrication (0.3 L/m^2^ h bar) compared to the PA TFC fabricated on the hydrophilic hydrogel (1 L/m^2^ h bar). This is due to the resistance of hydrophobic support to the penetration of PPD aqueous solution limiting the penetration of PPD into the pores that result in the formation of a thicker PA layer with lower permeability [12]. In addition, the effect of temperature on the performance of the fabricated thermo-responsive membranes was also studied by comparing the water permeability and rejection to NaCl at room temperature and beyond LCST of PNIPAAm hydrogel (45 °C) at a constant pressure of 1 bar. According to Table 5, by increasing the temperature, the water permeability increased by 6 times, whereas a slight decrease in the NaCl rejection was observed. At room temperature, water permeability and rejection were 0.3 L/m^2^ h bar and 32.4%, respectively, which changed to 1.8 L/m^2^ h bar and 27.9%, respectively, by increasing the temperature. At a temperature below LCST, hydration of the polymer network by water leads to the absorption of water and swelling of the hydrogel; hence, the resistance to water flow through the swollen pore-filling PNIPAAm hydrogel is high. However, at temperatures higher than the LCST of PNIPAAm, the hydrogel deswelling process occurs due to the breaking of hydrogen bonds; this leads to a phase separation that will open up channels through the hydrogel, which results in lower resistance to water flow [6,43,57,58,59,60,69]. Such temperature-dependent behavior of water flux has been studied and discussed in detail for PNIPAAm pore-filled PET track-etched membranes [40], which had been prepared by the same methodology used here. Wang et al. [69] also reported a higher flux for their membrane that was incorporated with PNIPAAm at a temperature above its LCST due to the swelling and deswelling of the membrane by temperature change. Guo et al. [70] developed a PNIPAAm-gelatin hydrogel membrane and observed the linear increase in water flux with the increasing temperature (from 25 to 45 °C) that was correlated to the change in the pore size of the membranes due to swelling and deswelling. The results of the present study indicate that the resistance of the support has a significant influence on the overall resistance of the TFC membrane. The surprising effect of the temperature-responsive wetting properties of the PNIPAAm in the support onto WCA of the PA TFC membrane (cf. Section 3.1.2) may be related to the fact that the barrier layer has nanofiltration characteristics as indicated by the only modest NaCl rejection. The slightly decreased NaCl rejection at the higher temperature would then be directly caused by the higher water permeability at the same pressure. Overall, the thermo-responsive behavior of developed TFC can be used for tuning and controlling membrane flux by temperature.

## 4. Conclusions

A novel thermo-responsive membrane with switchable hydrophilicity/-phobicity was successfully developed and employed as the support of a PA TFC membrane, replacing the conventional microporous membrane support. The characterizations of the fabricated membranes clearly indicated the attachment of PNIPAAm hydrogel into PET nonwoven and the successful PA fabrication on the PET-PNIPAAm membrane. The effect of temperature on the hydrophobic–hydrophilic behavior of the fabricated membranes was studied using water contact angle measurement above and below the LCST of PNIPAAm. The results showed thermo-responsive behavior of both the PET-PNIPAAm membrane and the PA TFC membrane, fabricated on top of the pore-filling PNIPAAm PET scaffold, due to the thermal sensitivity of integrated PNIPAAm hydrogel. Two versions of the newly fabricated PET-PNIPAAm-PA membrane, based on supports obtained by reactive pore-filling with or without PEG as a porogen, respectively, showed a rejection of 34% and 47% for NaCl, respectively, and flux of 1 L/m^2^ h at a low operating pressure of 1 bar. Finally, the impact of increasing temperature above LCST on the membrane performance was evaluated, which indicated a much higher flux with a slight decrease in NaCl rejection. The developed hydrogel-filled PET membranes in this study showed good potential to substitute the microporous membrane support layer (on a PET non-woven) used for conventional TFC membranes and yield smart thermo-responsive PA TFC membranes with improved efficiency and pronounced switchability.

## Figures and Tables

**Figure 1 polymers-14-03376-f001:**
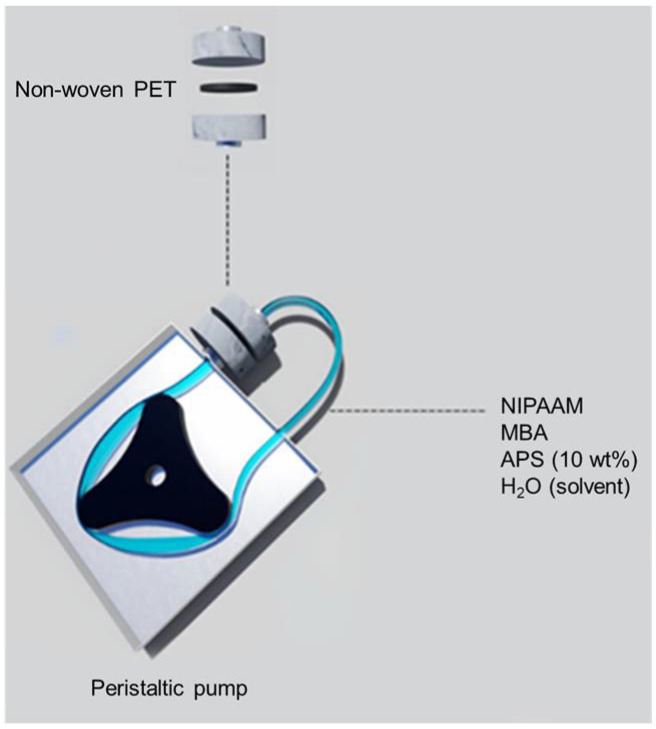
Circulation of reactive monomer solution through non-woven PET.

**Figure 2 polymers-14-03376-f002:**
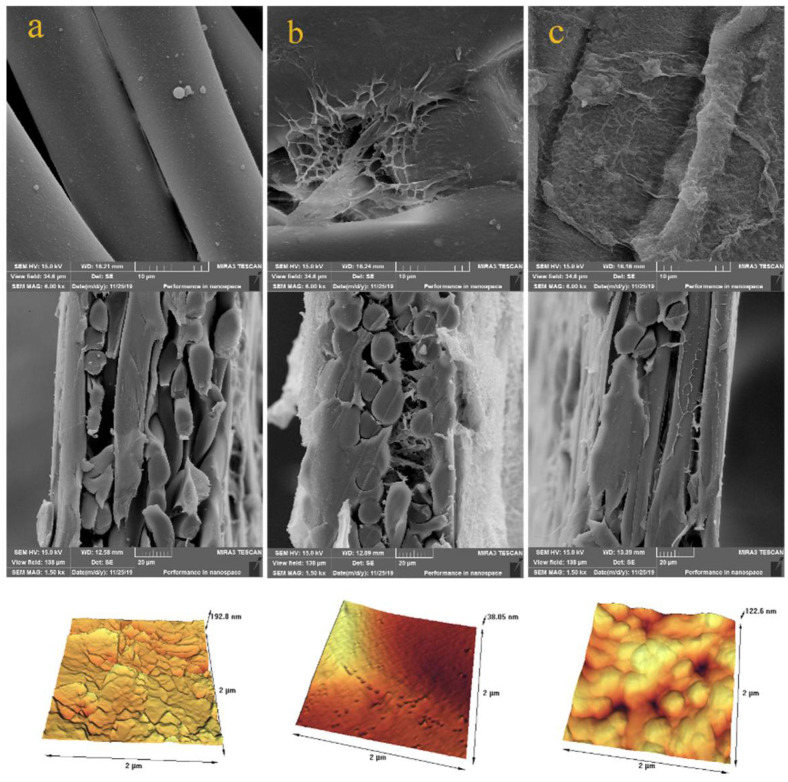
Surface and cross-sectional SEM and AFM images of (**a**) PET (**b**) PET-PNIPAAm (**c**) PET-PNIPAAm-PA membranes.

**Figure 3 polymers-14-03376-f003:**
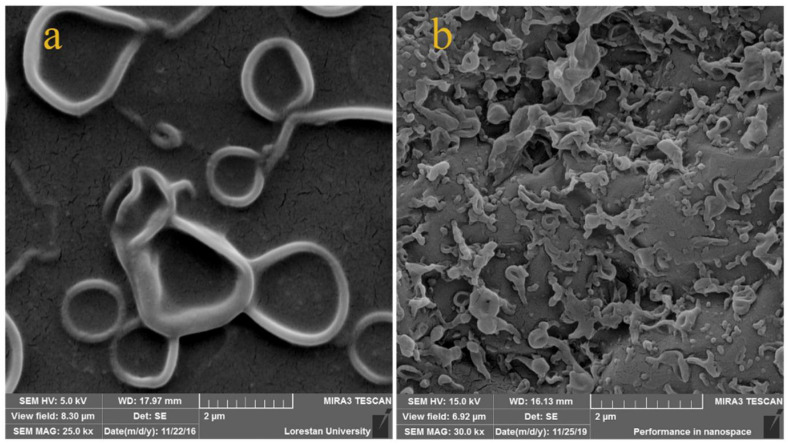
SEM micrographs of the surface of PA thin layer upon (**a**) conventional PES support and (**b**) PET-PNIPAAm support.

**Figure 4 polymers-14-03376-f004:**
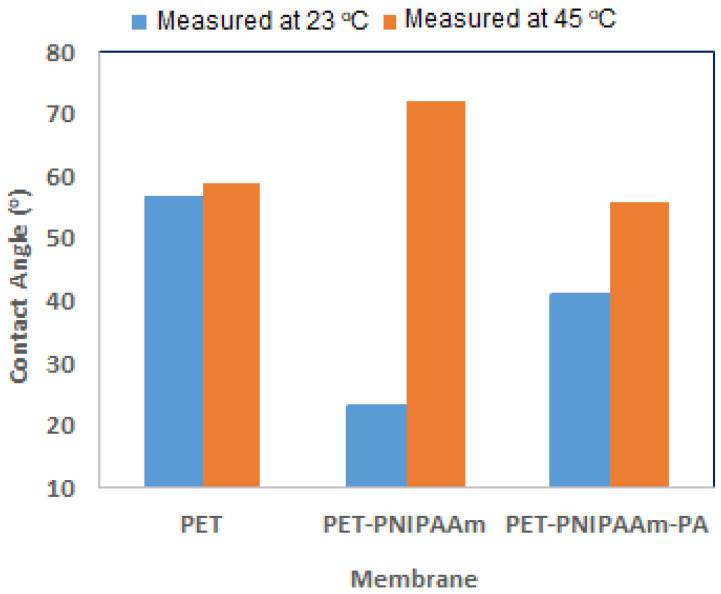
Water contact angle (WCA) values of membranes.

**Table 1 polymers-14-03376-t001:** Membrane surface roughness parameters from AFM.

Membrane	Ra (nm)	Rz (nm)	Rq (nm)
PET	3.5	23.3	24.2
PET-PNIPAAm	2.3	9.6	15.9
PET-PNIPAAm-PA	2.7	11.8	16.4

**Table 2 polymers-14-03376-t002:** Elements fraction (in atom and weight percent) from EDX for the different membranes.

Element	PET		PET-PNIPAAm		PET-PEG-PNIPAAm-PA	
	A %	W %	A %	W %	A %	W %
C	71.3	65.2	68.5	63.2	67.2	61.8
O	28.7	34.8	18.7	23	19.7	24.1
N	0	0	12.8	13.8	13.1	24.1

**Table 3 polymers-14-03376-t003:** Overview of membrane water permeability and salt rejection.

Membrane	Water Permeability (L/m^2^ h bar)	NaCl Rejection (%)
PET	16,920	0
PET-PNIPAAm	1.5	0
PET-PEG-PNIPAAm	37.4	0
PET-PNIPAAm-PA	1.0	33.8
PET-PEG-PNIPAAm-PA	1.0	47.0

**Table 4 polymers-14-03376-t004:** The performance comparison of PA TFC membranes reported in the literature with the ones established in this work.

Membrane	Water Permeability (L/m^2^ h bar)	NaCl Rejection (%) (bar)	Ref.
PA TFC/modified polyacrylonitrile	0.84	37.8 (5)	[65]
Commercial TFC-SR3	2.1	38 (10)	[66]
Commercial TFC-SR2	7.5	24 (10)	[66]
Modified PA TFC/PES	7.8	25.6 (6)	[67]
PA TFC/modified PES	11.4	31 (2)	[9]
PET-PNIPAAm-PA	1	33.8 (1)	This work
PET-PEG-PNIPAAm-PA	1	47.0 (1)	This work

**Table 5 polymers-14-03376-t005:** The effect of temperature on TFC PA membrane performance.

Membrane	Temperature	Water Permeability (L/m^2^ h bar)	NaCl Rejection (%)
PET-PNIPAAm-PA	Room temperature	1	33.8
PET-hydrophobic PNIPAAm-PA	Room temperature	0.3	32.4
PET-hydrophobic PNIPAAm-PA	45 °C	1.8	27.9

## Data Availability

The data presented in this study are available on request from the corresponding author.

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
