# Peer review of "Thermo-Responsive Hydrophilic Support for Polyamide Thin-Film Composite Membranes with Competitive Nanofiltration Performance"

_polymers, 2022, doi:10.3390/polym14163376_

Round 1

Reviewer 1 Report

In this manuscript, Poly(N-isopropylacrylamide) (PNIPAAm) was introduced to the polyethylene terephthalate (PET) nonwoven fabric to develop a stimuli-responsive polyamide thin film composite. Without using the microporous membrane interlayer, the thin film also possible be fabricated. The average water permeability, and an NaCl rejection at allow operating pressure were observed. However, there is very little relevant characterization contents and technologies, the comparison part between these composites with other materials are not enough to show the advantages of this material or method. Authors need major revisions to meet journal requirements.

1.     Authors must provide the FT-IR spectra to show the change in the composition.

2.     What are the real use cases of this membrane material and does the PNIPAAm coating come off due to the presence of water?

3.     Some mistakes in the subscript such as 45oc in the figures. Authors should very carefully check it through all the manuscript.

4.     What is the stress-strain curve of this membrane material, the authors need to test.

5.     Some related research about the porous polymers should be cited to compare with this PNIPAAm material. Nano Research, 2022: 15, 5556–5568.; Biomolecules, 2022, 12(5): 636.; Biomater. Sci., 2022, DOI: 10.1039/D2BM00719C.

Author Response

see uploaded file

Reviewer 2 Report

The work "Thermo-responsive hydrophilic support for polyamide thin-film composite membranes with competitive nanofiltration performance" has a very interesting and actual subject of the membrane and polymers research field.

The presented results are interesting and useful!

The methods and procedures are adequate!

The manuscript can be considered for publish, however some minor corrections were welcome:

- the authors must rewrite references on the Polymers journal standards.

- the SEM and AFM image must be more readable.

- the Figures 4 presentation must reconsidered as presentation form ( The graphic point by point is few close to reality).

for

Author Response

see uploaded file

Round 2

Reviewer 1 Report

possible accept